

# Exogenous melatonin promoted seed hypocotyl germination of *Paeonia ostia* 'Fengdan' characterized by regulating hormones and starches

Xia Li[1], Qi Sun[1], Qiang Shen[2], Chunlei Zhao[3], Fengzhen Chen[1], Yumei Liu[1], Guangcan Zhou[1], Xueqin Liu[1] and Xiaofei Kang[1]

[1] College of Agriculture and Bioengineering, Heze University, Heze, Shandong, China
[2] Heze Ruipu Peony Biotechnology Co., LTD, Heze, Shandong, China
[3] Heze Cunlei Horticulture Co., LTD, Heze, Shandong, China

Corresponding author
Xiaofei Kang, fay_ying@163.com

## ABSTRACT

**Background**. Seed hypocotyl germination signifies the initiation of the life cycle for plants and represents a critical stage that heavily influences subsequent plant growth and development. While previous studies have established the melatonin (MEL; N-acetyl-5-methoxytrytamine) effect to stimulate seed germination of some plants, its specific role in peony germination and underlying physiological mechanism have yet to be determined. This study aims to evaluate the MEL effect for the hypocotyl germination of peony seeds, further ascertain its physiological regulation factors.

**Methods**. In this work, seeds of *Paeonia ostia* 'Fengdan' were soaked into MEL solution at concentrations of 50, 100, 200, and 400 μM for 48 h and then germinated in darkness in incubators. Seeds immersed in distilled water without MEL for the same time were served as the control group.

**Results**. At concentrations of 100 and 200 μM, MEL treatments improved the rooting rate of peony seeds, while 400 μM inhibited the process. During seed germination, the 100 and 200 μM MEL treatments significantly reduced the starch concentration, and $\alpha$-amylase was the primary amylase involved in the action of melatonin. Additionally, compared to the control group, 100 μM MEL treatment significantly increased the $GA_3$ concentration and radicle thickness of seeds, but decreased ABA concentration. The promotion effect of 200 μM MEL pretreatment on $GA_1$ and $GA_7$ was the most pronounced, while $GA_4$ concentration was most significantly impacted by 50 μM and 100 μM MEL.

**Conclusion**. Correlation analysis established that 100 μM MEL pretreatment most effectively improved the rooting rate characterized by increasing $\alpha$-amylase activity to facilitate starch decomposition, boosting $GA_3$ levels, inhibiting ABA production to increase the relative ratio of $GA_3$ to ABA. Moreover, MEL increased radicle thickness of peony seeds correlating with promoting starch decomposition and enhancing the synthesis of $GA_1$ and $GA_7$.

## INTRODUCTION

Seed germination is the beginning stage of the plant life cycle, commencing with water uptake (suckering) and ending with radicle dewlap (radicle breakthrough into the seed coat or surrounding structures) (*Bewley et al., 2013*). This stage includes a series of physiological and metabolic processes, such as degradation of organic matter, protein synthesis and modification, and hormone synthesis and metabolism. It also represents the most critical stage and an adversity-sensitive period that determines plants' survival under unfavorable conditions. Peony is a traditional Chinese flower with increasing demand in both domestic and overseas markets. Although peonies can be propagated asexually by dividing and grafting, seed propagation is still widely used, for example, to breed seedlings for medicinal purposes and propagate rootstocks, and to select superior varieties from sown seedlings in crossbreeding (*Wang & Yuan, 2002*). In addition, there are many precious varieties and wild species with low seed germination rates under natural conditions. Take the wild *Paeonia ludlowii* as an example, the low seed germination rate makes it endangered in nature (*Jing et al., 1995*). Therefore, it is of great interest to study how to improve the germination rate of peony seeds. *Paeonia ostia* 'Fengdan' is a renowned ornamental, oil and medicinal peony, and it is also an excellent rootstock for ornamental peony grafting, which has an important position in the industrialized production of peony seedlings (*Cheng & Du, 2008*). Sowing is the traditional way to propagate 'Fengdan', but the seeds have double dormancy characteristics. Initially, fresh seeds need to go through a period of after-ripening to break the hypocotyl dormancy, leading to extended germination times and relatively low germination rates (*Cheng & Du, 2008*; *Li et al., 2022*).

Seed priming can increase seed germination rate and germination uniformity by reducing suckering time (*Brocklehurst & Dearman, 1983*), increasing activation of pre-emergence enzymes, metabolite production (*Hussain et al., 2016*), repairing damaged DNA (*Farooq et al., 2009*) and regulating osmosis. It has been shown that gibberellin (GA) treatment can promote peony seed germination (*Ren et al., 2016*). For example, pretreatment with 300 mg L$^{-1}$ gibberellin (GA$_3$) for 24 h and germination at 20 °C enhanced the rooting of 'Fengdan seeds' compared to untreated seeds (*Ren et al., 2016*). About 75.56 ± 8.39% germination rate of *Suaeda glauca* black seeds was achieved in the presence of 1 mM GA$_3$ conditions after 30 days of incubation at 20 °C/10 °C with 12 h photoperiod conditions, while approximately 16.7% germination was obtained for seeds treated without GA$_3$ (*Wang et al., 2024a*; *Wang et al., 2024b*). Melatonin was first discovered in vascular plants by researchers in 1995 (*Dubbels et al., 1995*; *Hattori et al., 1995*), and tryptophan, its precursor, is a biomolecule considered to have possible hormonal activity (*Ludwing-Müller & Lüthen, 2015*; *Arnao & Hernández-Ruiz, 2018*). Many studies have explained the regulation of plant biological functions by melatonin, such as growth of explants, sprouting of shoots, roots and seeds (*Kołodziejczyk, Kaźmierczak & Posmyk, 2021*), as well as promoting seed germination under abiotic stress (*Wang et al., 2024a*; *Wang et al., 2024b*; *Tian et al., 2024*). Melatonin has a similar structure and synthesis pathway to indoleacetic acid (IAA) and is therefore thought to have similar functions to IAA (*Murch & Saxena, 2002*), such as the ability to induce stem and root growth, stimulate root production, and produce new lateral

branches and adventitious roots. Exogenous low concentrations of melatonin significantly promote seed germination in a wide range of plants (*Hernándes-Ruiz, Cano & Arnao, 2005*). *Xiao et al. (2019)* obtained that 20 µM melatonin treatment optimally promoted cotton (*Gossypium hirsutum* L.) seed germination, increasing germination potential, germination rate and final fresh weight by 16.67%, 12.30% and 4.81%, respectively. *Simlat et al. (2018)* found that melatonin promoted the germination of *Stevia rebaudiana* Bertoni seeds and played a crucial role in the development of stevia seedlings: fresh weight and the number of leaves of plants in the 20 µM MEL-treated group were significantly increased; the height of plants in the 5 µM group was improved; and the best root development was observed in the 500 µM group along with increased sugar and phenolic concentrations, as well as catalase and peroxidase activities. *Dong et al. (2021)* concluded that appropriate exogenous MEL can affect some regulatory pathways in *Zoysia japonica* Steud., resulting in improved the antioxidant capacity and reduced the content of cytokinin (CTK), GA, and ABA in seeds. In addition, melatonin has been found to regulate the germination and foster the production of antioxidant nutrients in aged legumes (*Yu et al., 2021a*), offering a potential novel approach to enhance the germination rate of aged seeds.

While melatonin pretreatment has been observed to influence the seed germination of various plants, its regulatory impact on peony seed germination remains unexplored. In the current study, 'Fengdan' (*Paeonia ostia*) seeds were subjected to different concentrations of melatonin solution to ascertain the effects of melatonin on the hypocotyl germination. Furthermore, the concentrations of GA, ABA, starch, sucrose, free fatty acids, and activities of related enzymes in the seeds were determined to investigate the primary regulatory factors of melatonin on peony seeds. This study aims to investigate the effectiveness of melatonin in promoting peony seed germination and identify its physiological regulatory factors.

## MATERIALS & METHODS

### Plant materials

The seeds of 'Fengdan' (*Paeonia ostia*) were collected from peony planting base in Heze, Shandong Province (35°31′N, 115°43′), on August 15, 2022, with a thousand-seed weight of 448.54 g. The seeds were stored at 4 °C and 70% relative humidity after harvest.

### Germination experiment

The germination experiments were initiated on September 30, 2022. Soak the full-bodied 'Fengdan' seeds in distilled water for 1 h, select the ones that sink in the water and put them into 0.1% formaldehyde solution to sterilize for 30 min, rinse with distilled water for 5 times, and then dry the surface water in a cool place. The same number of sterilized seeds was taken into MEL solution at concentrations of 50, 100, 200, and 400 µM for 48 h, respectively. Seeds immersed in distilled water without MEL for equivalent duration were used as the control group. After soaking, the seeds were evenly placed in a petri dish (diameter 20 cm) with two layers of filter paper under the seeds and an equal amount of distilled water. Each group (M0, M50, M100, M200, M400) was comprised of three repetitions, with 60 seeds in each repetition. Then, the seeds were germinated in darkness,

at 20 °C and relative humidity of 65% in an incubator. Observe rooting daily and record the time of rooting. When the germination rate of each group remained unchanged for three consecutive days, the germinated seeds were taken for physiological indicators. The standard for germination was defined by the splitting of the seed coat and the emergence of a white radicle.

## Determination of morphological and physiological indexes

Rooting rate = (number of rooted seeds/total number of seeds) × 100%. The diameter of radicle was measured with an electronic vernier caliper.

Starch content was estimated according to the acid hydrolysis-DNS colorimetric method (*Dai et al., 2013*). Seeds power (0.2 g) were prepared and subjected to fat removal using 25 mL of petroleum ether and soluble carbohydrate extraction using 85% ethanol. To the residue, five mL of water and 1.6 mL of 1:1 hydrochloric acid were added, and the mixture was dispersed and hydrolyzed in a boiling water bath for 2 h. After hydrolysis, the sample was immediately cooled. The hydrolysate was then transferred to a 25 mL volumetric flask, and the centrifuge tube was washed with a small amount of water. The washing solution was combined with the hydrolysate in the volumetric flask, and one drop of methyl red indicator was added. The pH was adjusted to 7 by adding a 400 g/L sodium hydroxide solution until the mixture turned yellow, followed by 1:1 hydrochloric acid until it just turned red. Subsequently, one mL of a 200 g/L lead acetate solution was added, shaken well, and left to stand for 10 min. To remove excess lead, one mL of a 100 g/L sodium sulfate solution was added, and the volume was adjusted to the mark with distilled water, followed by thorough mixing. The sample was then filtered, with the initial filtrate discarded, and the final filtrate was analyzed using a T6 ultraviolet–visible spectrophotometer (Persee, Beijing, China).

Peony seeds (0.5 g) were homogenized in 10 mL of distilled water and allowed to sit at room temperature for 15 min, with shaking every 5 min to ensure full extraction. The mixture was then centrifuged at 6,000 g for 10 min, and the supernatant was collected and adjusted to a final volume of 10 mL. This constituted the amylase stock solution. Subsequently, the original amylase solution was diluted by a factor of five. The concentrations of $\alpha$-amylase and $\beta$-amylase were determined using a T6 UV-visible spectrophotometer, following the methodology described by *Wang (2015)*.

Free fatty acids concentration and lipase activity of peony seeds were assayed by the Free Fatty Acid Kit (BC0590; Solarbio) and the Lipase Activity Assay Kit (BC2340; Beijing Solarbio Science & Technology, Beijing, China) according to the manufacturer's instructions. Sucrose content was determined with reference to the method of *Zhang & Zhai (2003)*. Seed sample (0.1 g) and four mL 80% ethanol solution were added into 10 mL centrifuge tube successively. The sample was soaked in an 80 °C water bath for 40 min and centrifuged to separate the supernatant. Then, the residue was extracted twice with 2 mL of 80% ethanol each time. The supernatants were combined and 10 mg of activated carbon was added to decolorize at 80 °C for 30 min. Finally, the volume was adjusted to 10 mL, filter, and the content was determined using a T6 UV-visible spectrophotometer.

GA and ABA concentrations were determined by liquid chromatography (Agilent 1260 HPLC in tandem with a G6420A mass spectrometer), and the method was referred to *Pan, Welti & Wang (2010)*. Peony seeds powder (0.8 g) and one mL of extraction solution (isopropyl alcohol: ultra-pure water: hydrochloric acid = 1000:500:1) was added to the test tube successively, oscillated at 4 °C for 30 min in the dark. Following this, 2 mL of dichloromethane was introduced and oscillated at 4 °C for another 30 min in the dark. The mixture was then centrifuged at 4 °C at 13,000 rpm for 5 min. The lower organic phase was collected into a centrifuge tube containing 0.2 g of anhydrous magnesium sulfate, mixed on a vortex for 1 min, and centrifuged. The supernatant was transferred to a new 10 mL centrifuge tube, concentrated to dryness at a temperature below 30 °C using a centrifuge concentrator, redissolved in 0.2 mL of 0.1% formic acid-methanol solution, and filtered through a 0.22 μm filter membrane. The sample was then subjected to HPLC-MS/MS analysis.

## Data analysis

Graphing and statistical analysis of morphological and physiological indexes data was done using GraphPad Prism v.9.0 software. The overall treatments were determined by one-way analysis of variance (ANOVA) followed by Tukey's LSD post-hoc test. The significance of the observed variables among different treatments was estimated at $P < 0.05$. Pearson's correlation was employed to assess the relationship among MEL concentration, germination rate, radicle thickness and physiological indexes, with a correlation coefficient threshold of $|\rho| > 0.7$ and a significance level of $P < 0.01$.

## RESULTS

### Effect of MEL treatment on rooting rate and radicle diameter of peony seeds

Our study found that although MEL treatment did not accelerate the rooting speed of peony seeds, it did enhance the rooting rate, radicle thickness, and uniformity of the seed population. As shown in Fig. 1, M 100 and M 200 significantly promoted the rooting rate of 'Fengdan' seeds, with rooting rates 1.41 and 1.25 times that of the control, respectively, while M400 inhibited rooting. The 100 μM and 200 μM MEL treatments also significantly increased the radicle thickness of 'Fengdan' seeds, which was 1.28 and 1.25 times that of the control, respectively.

### Effect of MEL treatment on carbohydrates and FFA metabolism of peony seeds

As starch is the main nutrient of seeds, changes in starch content as well as amylase activity can reflect the influence of MEL on seed germination. Figure 2A showed that M100 and M200 decreased the starch concentration of seeds to 74.3% and 71.8% of the control, respectively. The $\alpha$-amylase activity was highest for M200, followed by M100, which was 2.03 and 1.77 times higher than the control, respectively (Fig. 2B). The activity of $\beta$-amylase

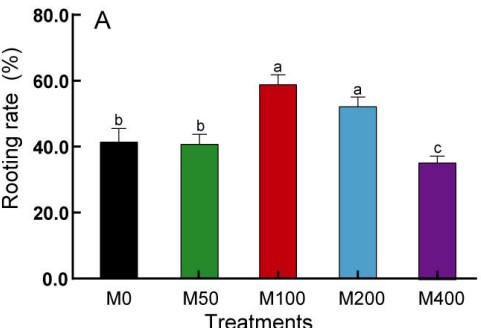
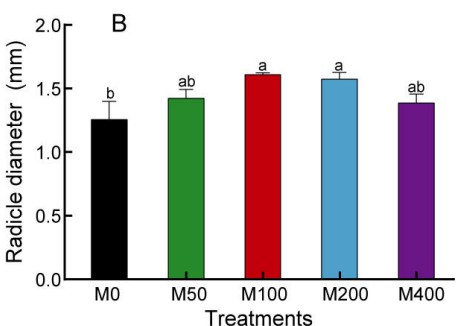

**Figure 1** **Effects of melatonin treatment on seed germination rate (A) and radicle diameter (B).** Note: Different letters mean significantly at $p < 0.05$ level applying Tukey's test. M0, distilled water pretreatment, no exogenous melatonin; M50, 50 $\mu$M melatonin solution pretreatment; M100, 100 $\mu$M melatonin solution pretreatment; M200, 200 $\mu$M melatonin solution pretreatment; M400, 400 $\mu$M melatonin solution pretreatment.

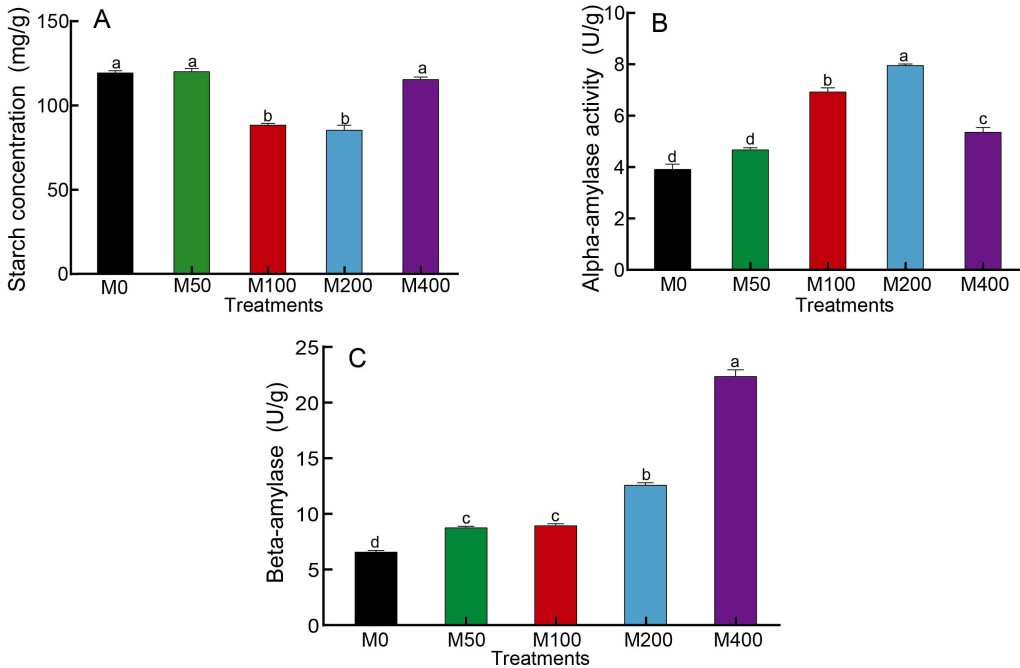

**Figure 2** **Effects of melatonin treatment on starch concentration (A), $\alpha$-amylase (B) and $\beta$-amylase (C) activities of peony seeds.**

was highest in M400, being 3.37 times that of the control, and other MEL treated groups were also noticeably higher than the control (Fig. 2C).

It was demonstrated that different concentrations of MEL treatments significantly increased the free fatty acid concentration in seeds and enhanced fat metabolism (Fig. 3A). Among them, M200 was the highest, which was 3.99 times that of the control. The changes in lipase activity were different from free fatty acids: M50 was the highest, M100, M200

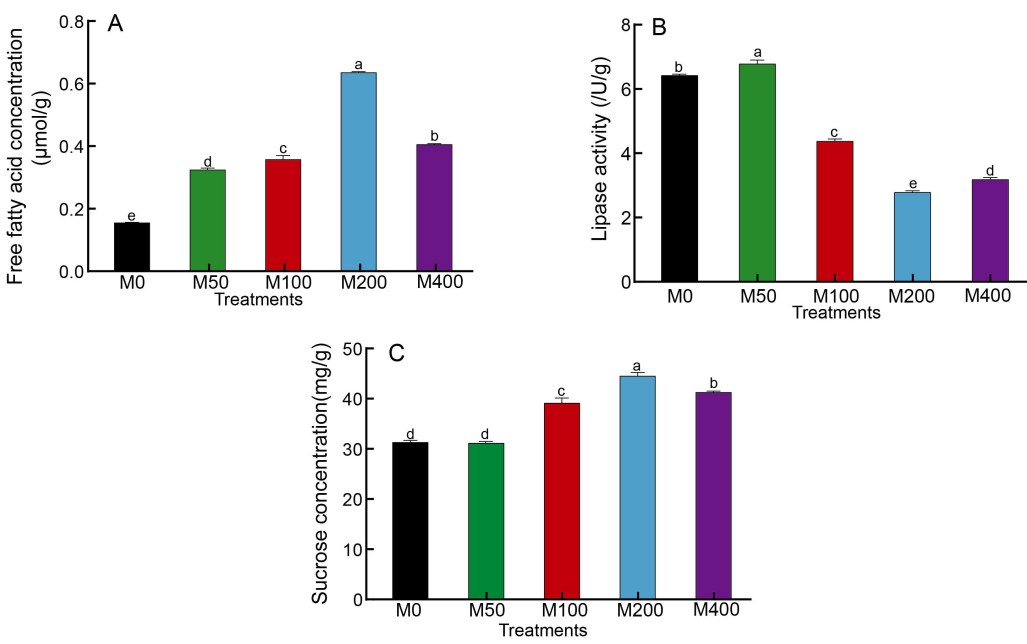

**Figure 3** Effects of melatonin treatment on free fatty acid concentration (A), lipase activity (B) and sucrose concentration (C) of peony seeds.

and M400 were significantly lower than the control, and M200 was the lowest at 43.81% of the control (Fig. 3B). The effect of MEL treatment on the concentration of sucrose in seeds was similar to that of free fatty acids. M100, M200 and M400 were all markedly higher than the control, with 1.25, 1.42 and 1.31 times higher than the control, respectively (Fig. 3C).

## Effect of MEL treatment on GA and ABA concentration of peony seeds

GA is the key hormone for breaking hypocotyl dormancy in plant seeds, especially $GA_3$, which is commonly used to promote seed germination. Figure 4 revealed that MEL treatment resulted in a notable increase in seed $GA_1$ and $GA_7$ concentrations compared to the control, but different concentrations of MEL had different effects. M100 and M200 better promoted the accumulation of $GA_1$, which was 1.59 and 1.62 times that of the control, respectively (Fig. 4A). $GA_7$ was not detected in the control, whereas it was detected in all MEL-treated groups, with the highest amount in the M200 (Fig. 4B). M50 and M100 had the highest $GA_4$ concentrations, 2.31 and 2.1 times that of the control, respectively (Fig. 4C). The highest $GA_3$ content was found in M100, being 1.57 times of the control (Fig. 4D).

Seed germination is usually inhibited by the presence of ABA, which together with GA regulates the number of seeds germinating. As analyzed in Fig. 4E, the ABA concentration of M100 was only 75.87% of the control, showing the greatest decrease. Figure 4F reveals the differences in $GA_3$/ABA between groups, with M100 being the highest at 2.07 times that of the control, followed by M200, 1.44 times the control.
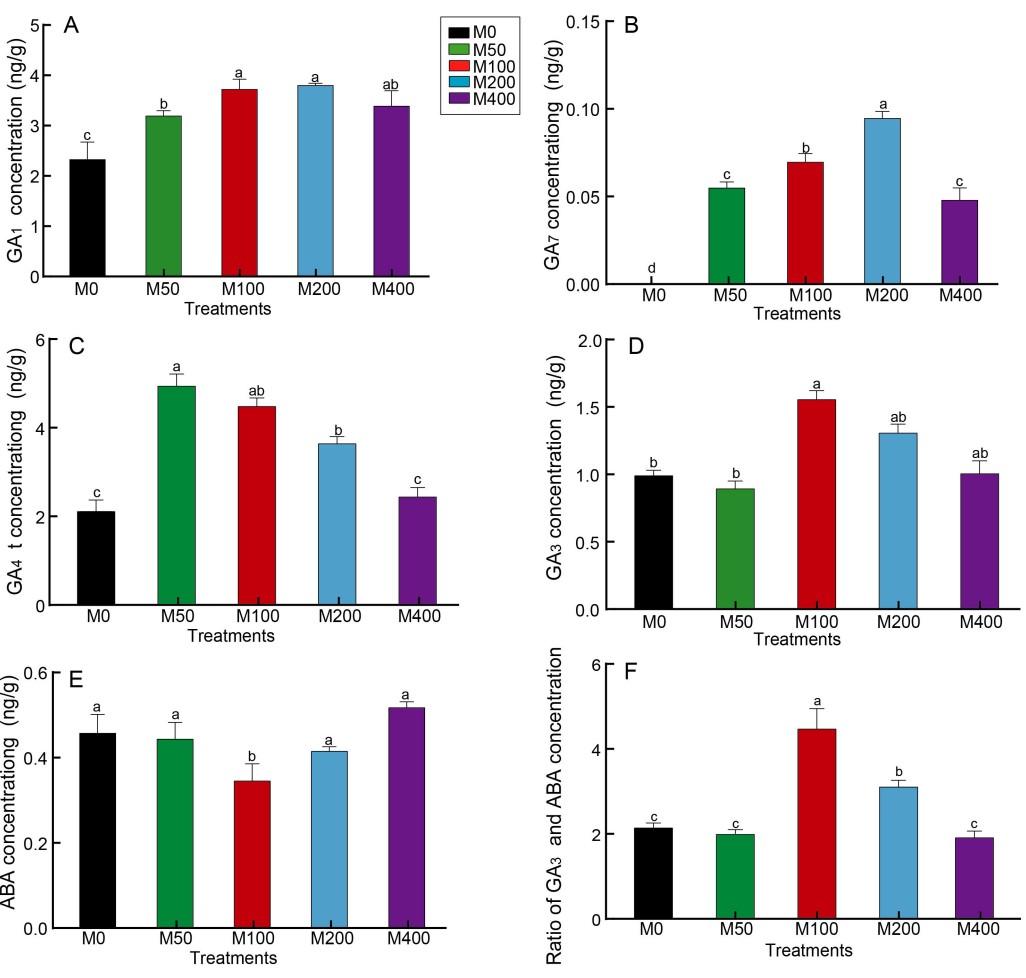

**Figure 4   Effects of melatonin treatment on GA$_1$ concentration (A), GA$_7$ concentration (B), GA$_4$ concentration (C), GA$_3$ concentration (D), ABA (E) concentration and GA$_3$/ABA (F) of peony seeds.**

## Correlation between rooting rate, radicle diameter and various physiological indicators

According to the correlation analysis between physiological indexes and rooting rate (Fig. 5), the rooting rate of 'Fengdan' was positively correlated with GA$_3$ ($p < 0.05$) and GA$_3$/ABA ($p < 0.01$), and significantly negatively correlated with starch ($p < 0.05$) and ABA ($p < 0.01$). In addition, radicle coarseness exhibited a significant positive correlation with GA$_1$ and GA$_7$ ($p < 0.05$), and a negative correlation with starch concentration ($p < 0.01$).

## DISCUSSION

After seeds swell and begin to sprout, the stored nutrients are gradually degraded and the relative catabolic enzyme activities start to increase, providing energy and components for root growth. Starch is the main storage form of sugar. At the initial stage of seed germination, the degradation of starch requires the participation of many enzymes,

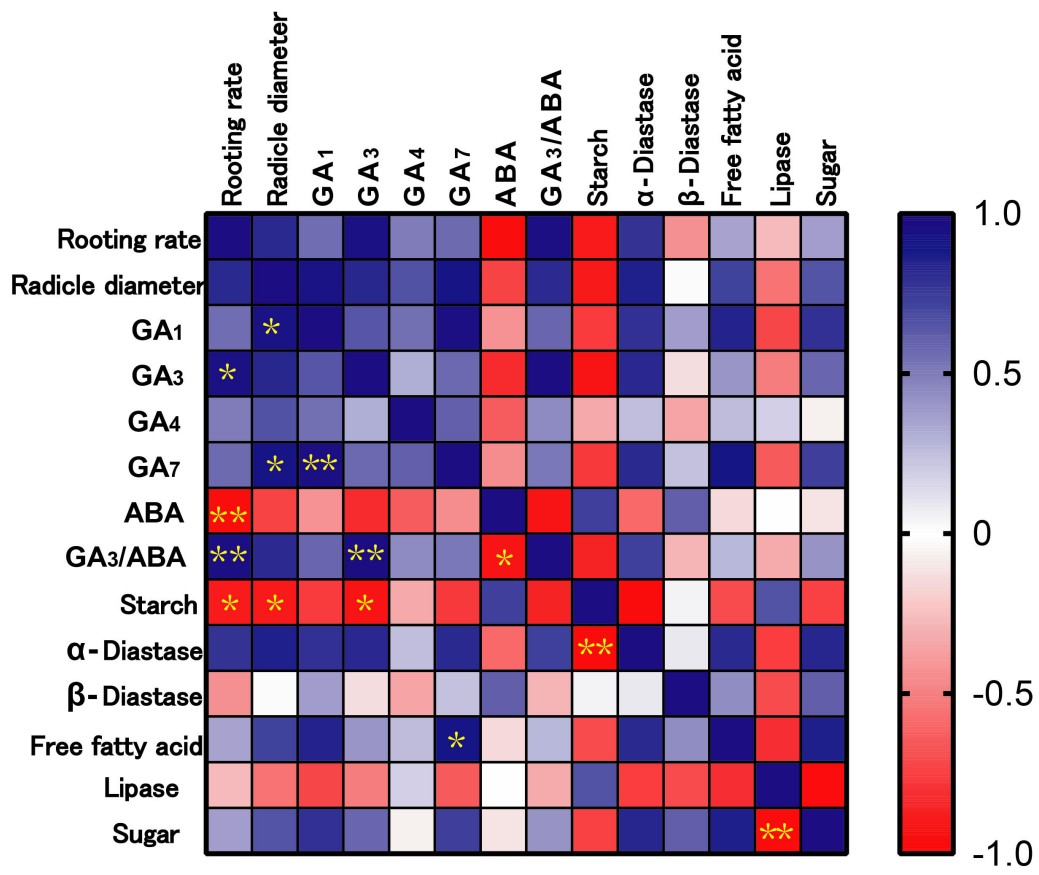

**Figure 5** Correlation analysis among rooting rate, radicle diameter and physiological indexes. Note: * mean significant correlation at $p < 0.05$, ** mean significant correlation at $p < 0.01$.

including $\alpha$-amylase, $\beta$-amylase, amylopectin, dextran phosphorylase and limit-dextrinase. Among these, $\alpha$-amylase activity activation may be induced by gibberellins (*Beck & Ziegler, 1989*). During the early stages of seed germination, $\alpha$-amylase hydrolyzes starch to small molecular sugars, and its activity is also correlated with the rate of seed germination as well as seedling growth and development. It is thought to be the enzyme that initially drives the degradation of starch granules to soluble, small molecule dextrans (*Smith, Zeeman & Smith, 2005*). $\beta$-amylase is an exoamylase, which hydrolyzes starch into maltose. In short, starch is broken down into sugars by the enzyme amylase to provide the energy needed for seed germination. Studies have shown that melatonin can promote the activities of $\alpha$-amylase and $\beta$-amylase in cotton seeds, regulate energy production and reduce the inhibition of sodium chloride stress on seed germination (*Chen et al., 2020*; *Chen et al., 2021*). The results of our study were consistent with previous findings. MEL treatment was able to affect the activity of amylase, but different amylases responded to melatonin at different concentrations, with $\alpha$-amylase activity being the highest under 200 μM melatonin treatment and $\beta$-amylase activity being the highest under 400 μM melatonin treatment. Correlation analysis revealed that $\alpha$-amylase was significantly negatively correlated with
starch concentration ($p < 0.01$), and the starch concentration was negatively correlated with seed rooting rate and radicle diameter ($p < 0.05$). These findings indicated that melatonin treatment can effectively enhance $\alpha$-amylase activity, promote the decomposition of starch, and facilitate seed hypocotyl germination.

As the seeds of oilseed crops germinate, their lipids degrade and the free fatty acid as well as sucrose content increases (*Tai & Zeiger, 2015*). This process commences in the oleosomes, where triacylglycerol undergoes hydrolysis to form free fatty acids. Subsequently, these fatty acids are oxidized to acetyl CoA, and further metabolized in the glyoxylate cycle and cytoplasm to produce succinic acid. The succinic acid is then transformed into malic acid in the mitochondria, and finally converted to glucose through gluconeogenesis in the cytoplasm, ultimately forming sucrose. In most oil-storing seeds, approximately 30% of the acetyl CoA is utilized for respiratory metabolism to provide energy for cells, while the remaining portion is converted to sucrose (*Tai & Zeiger, 2015*). And lipase activity plays a relevant role in the mobilization of reserves (stored as TAGs) during oilseed germination (*Beisson et al., 2000*; *Graham, 2008*). For example, during jojoba (*Simmondsia chinensis*) germination, the first step of wax ester mobilization is catalyzed by lipases (*Kawiński et al., 2021*). 'Fengdan' seeds are used to extract peony oil in China. Therefore, three main indexes including free fatty acids, lipase and sucrose were detected in this work. The changes in sucrose concentration and free fatty acids showed similar patterns, with melatonin treatments of 100 µM, 200 µM, and 400 µM exhibiting significantly higher levels compared to the control ($p < 0.05$). This indicated that melatonin treatment accelerated the breakdown of lipids and enhanced sucrose production. This result was consistent with those of *Simlat et al. (2018)*, who also confirmed that the concentration of sugar increased under the 500 µM MEL treatment to stevia (*Stevia rebaudiana* Bertoni) seeds. The alterations in lipase activity diverge from those observed in sucrose and free fatty acids. The 50 µM melatonin treatment yielded the highest lipase activity, while the 100 µM, 200 µM, and 400 µM melatonin treatments resulted in significantly lower activity compared to the control ($p < 0.05$). This possibly owed to variations in lipase activity resulting from different phases of seeds germination under different treatments at the time of sampling. Lipase activity was detectable prior to seed imbibition and increased during germination (*Patui et al., 2014*). Since there are few studies on the changes of lipase during seed germination, the specific reasons need to be further analyzed by dynamic changes in the future.

GA and ABA are plant hormones that play pivotal roles in seed germination and early seedling establishment (*Shu et al., 2018*; *Ahammed et al., 2020*). ABA is a universal abiotic stress hormone with a positive response to abiotic stress, while GA acts as a plant growth regulator that promotes seed germination under stress conditions (*Shu et al., 2018*). Studies have demonstrated that soaking seeds in a melatonin solution can enhance the germination rate of cucumber (*Cucumis sativus* L.), cotton (*Gossypium hirsutum* L.), alfalfa (*Medicago sativa*), and various aged soybean seeds under salt stress (*Zhang et al., 2014*; *Chen et al., 2021*; *Yu et al., 2021b*). The mechanism may be attributed to increased melatonin can enhance the antioxidant activity of seeds, thereby improving seed vitality and germination rates. Furthermore, melatonin treatment may regulate the anabolism of the plant hormones

GA and ABA catabolism, promoting seed germination by up-regulating the synthesis of $GA_3$ in seeds while also promoting the expression of ABA metabolism gene *CYP707A* family genes and down-regulating its synthesis gene *NCED* to accelerate ABA catabolism. Similar results were reported by *Bai et al. (2020)*, where the pretreatment of seeds with 100 μM melatonin alleviated drought stress and increased the germination rate of cotton seeds (*Gossypium hirsutum* L.) by influencing ABA and $GA_3$ contents.

In this study, consistent with previous findings, exogenous melatonin treatment resulted in a significant increase in the concentration of GAs under non-stress conditions. It was observed that different concentrations of melatonin had varying effects on different types of GA. The 100 μM MEL treatment demonstrated the highest effect of $GA_3$, simultaneously suppressed ABA synthesis. Correlation analysis revealed a positive relationship between the rooting rate and $GA_3$ ($p < 0.05$) and $GA_3$/ABA ($p < 0.01$), as well as a negative correlation with ABA ($p < 0.01$), indicating that melatonin treatment may improve rooting rate by enhancing $GA_3$ and inhibit ABA synthesis. *Xiao et al. (2019)* also confirmed the similar mechanism that 20 μM melatonin treatment reduced ABA content in cotton seeds by 42.13–51.68%, while increased $GA_3$ content to about 1.7–2.5 times that of seeds germinated without melatonin.

## CONCLUSIONS

In summary, 100 μM exogenous MEL pretreatment increased the rooting rate and radicle thickness of 'Fengdan' seeds. MEL treatment regulated seed hypocotyl germination characterized by promoting starch decomposition within the seed, promoting $GA_3$ synthesis, inhibiting ABA synthesis, and increasing the value of $GA_3$/ABA. In addition, MEL treatment increased radicle thickness significantly correlated with promoting starch catabolism and the synthesis of $GA_1$ and $GA_7$. Our research highlights the efficacy of melatonin in promoting peony seed hypocotyl germination and its physiological regulatory characteristics through the correlation analysis of morphological and physiological indexes.

## ACKNOWLEDGEMENTS

The authors express their gratitude to Heze Ruipu Peony Biotechnology Co., LTD and Heze Cunlei Horticulture Co., LTD for generously supplying the peony seeds.

### Funding

This study was supported by the Fundamental Research Funds for Heze University (Grant No. XY21BS28; XY22BS23), the Yellow River Basin Ecological Protection and High-quality Development Collaborative Innovation Center of Shandong Province (2022), the Key Laboratory of Natural Product Functional Food and Reproductive Nutrition of Heze, and the Food and Reproductive Nutrition Functional Food Technology Innovation Center of Heze. The funders had no role in study design, data collection and analysis, decision to publish, or preparation of the manuscript.

## Grant Disclosures

The following grant information was disclosed by the authors:

Fundamental Research Funds for Heze University: XY21BS28, XY22BS23.

Yellow River Basin Ecological Protection and High-quality Development Collaborative Innovation Center of Shandong Province (2022).

Key Laboratory of Natural Product Functional Food and Reproductive Nutrition of Heze.

Food and Reproductive Nutrition Functional Food Technology Innovation Center of Heze.

## Competing Interests

Qiang Shen is employed by HezeRuipu Peony Biotechnology Co., LTD. Chunlei Zhao is employed by HezeCunlei Horticulture Co., LTD. The authors declare there are no competing interests.

## Author Contributions

- Xia Li conceived and designed the experiments, authored or reviewed drafts of the article, and approved the final draft.
- Qi Sun performed the experiments, prepared figures and/or tables, and approved the final draft.
- Qiang Shen performed the experiments, prepared figures and/or tables, and approved the final draft.
- Chunlei Zhao performed the experiments, prepared figures and/or tables, and approved the final draft.
- Fengzhen Chen conceived and designed the experiments, prepared figures and/or tables, authored or reviewed drafts of the article, and approved the final draft.
- Yumei Liu analyzed the data, prepared figures and/or tables, and approved the final draft.
- Guangcan Zhou analyzed the data, prepared figures and/or tables, and approved the final draft.
- Xueqin Liu conceived and designed the experiments, prepared figures and/or tables, and approved the final draft.
- Xiaofei Kang analyzed the data, authored or reviewed drafts of the article, and approved the final draft.

## Data Availability

The raw measurements are available in the Supplementary Files. The raw data shows 100 $\mu$M exogenous MEL pretreatment increased seed germination by promoting starch decomposition within the seed, promoting GA3 synthesis, inhibiting ABA synthesis, and increasing the value of GA3/ABA; increased radicle thickness by promoting starch catabolism and the synthesis of GA1 and GA7.

## Supplemental Information

Supplemental information for this article can be found online at http://dx.doi.org/10.7717/peerj.18038#supplemental-information.

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
