# Peer review of "Exogenous melatonin promoted seed hypocotyl germination of Paeonia ostia ‘Fengdan’ characterized by regulating hormones and starches"

_PeerJ, doi:10.7717/peerj.18038_

## Round 0.1 · original submission · Major Revisions

Please revise the manuscript according to reviewers' comments.To allow the audience to replicate this study, a more detailed materials and methods is needed. The discussion section also requires improvement.

Reviewer 1 ·

Basic reporting

No comment

Experimental design

No comment

Validity of the findings

No comment

Additional comments

This study explored the effects of melatonin on the physiological mechanisms of peony seed germination and has original innovation. Overall, this manuscript has clear English, sufficient background, and clear figures. But in my opinion, further improvement is needed.
1. A clear purpose needs to be added to the abstract.
2. References are needed. Lines 73-75, 79-82.
3. Please provide the longitude and latitude. Line 126
4. The experimental method description of this study is too simplistic, and the measurement method needs to include detailed steps and provide information on the required reagents and instruments
5. The sentences should be put into discussion. Lines 174-175, 184-185.
6. Please check figure. 1 A about the letters of statistical difference.
7. Most of the discussion in this manuscript is focused on repeating the results and does not compare or discuss with other related studies. I suggest adding more comparisons and discussions on relevant research.

Annotated reviews are not available for download in order to protect the identity of reviewers who chose to remain anonymous.

Reviewer 2 ·

Basic reporting

This is a very interesting article. The literature review justifies the choice of species for the study and provides insight into the role of melatonin during germination. The literature is well chosen. The chapter needs some additions, comments in the text of the manuscript. I propose to take the discussion a little further. Chart 2B should be corrected.

Experimental design

Materials and methods chapter needs improvement, comments in the text. In my opinion, when the authors refer to literature in Chinese, the methods should be described in detail. This will enable others to verify and use it in their research.

Validity of the findings

Thank you for sending the raw data. This made the work easier and allowed me to verify the graphs and description of the results. I left comments in the manuscript.

Annotated reviews are not available for download in order to protect the identity of reviewers who chose to remain anonymous.

Reviewer 3 ·

Basic reporting

no comment

Experimental design

no comment

Validity of the findings

no comment

Additional comments

Given that the germination potential variability, 180 seeds seem to be a large population to conduct the experiments to minimize the background variability. The data of this paper is convincing and straightforward. However, a couple of concerns are raised based on the data (see below) and lack of interpretation on the presented data also makes the paper very descriptive.

Major concerns:

For example, when authors detect starch and amylase activity in Figure 2 and corresponding texts, do authors see other major carbohydrate changes? Are starch and amylase “the big elephant in the room” during the germination process?

Line 184-192: figure 3A-B and text, the function of lipase is breaking down fat into free fatty acids, when low lipase activity is detected, amounts of fatty acids are also low generally, at least overall trend. However, Figure 3A and B don't match and correlate. The authors also try to claim the conversion relationship between lipid and sucrose, but the evidence is very weak due to lack of key enzyme results/data here.

---

## Round 0.2 · Minor Revisions

# Dear authors, please address the following concerns:

The title and conclusions are not justified. The paper describes a correlational study. This does not show that MEL acts via regulation of hormone and starch levels, merely that they are correlated.

Another issue is that the methods are unclear. The methods describes that when more than half of the seeds had germinated with root length 1-3 cm then they were sampled for physiology. What was sampled? The germinated seed? The mixture of germinated and germinated seed? Also, since three of the five treatments had rooting rate of 40% or less, how was it possible to sample when > 50% of the seed had roots > 1 cm?

Thanks,

Reviewer 1 ·

Basic reporting

no comment

Experimental design

no comment

Validity of the findings

no comment

Additional comments

This manuscript has addressed most of the issues and is recommended for acceptance.

Reviewer 3 ·

Basic reporting

N/A

Experimental design

N/A

Validity of the findings

N/A

Additional comments

I have checked the response letter, revised manuscript, and raw data. I think the authors roughly address most of my comments as I asked before.

I have no additional comments.

---

## Round 0.3 · accepted · Accept

Thanks for authors' response and revision. I believe authors have sufficiently addressed our section editor's comments.